# Cystic Fibrosis Clinical Isolates of *Aspergillus fumigatus* Induce Similar Muco-inflammatory Responses in Primary Airway Epithelial Cells

**DOI:** 10.3390/pathogens10081020

**Published:** 2021-08-13

**Authors:** Samantha A. McLean, Leilani Cullen, Dianne J. Gardam, Craig J. Schofield, Daniel R. Laucirica, Erika N. Sutanto, Kak-Ming Ling, Stephen M. Stick, Christopher S. Peacock, Anthony Kicic, Luke W. Garratt

**Affiliations:** 1Wal-Yan Respiratory Research Centre, Telethon Kids Institute, University of Western Australia, Crawley 6009, Australia; Samantha.McLean@telethonkids.org.au (S.A.M.); craig.schofield@telethonkids.org.au (C.J.S.); daniel.laucirica@telethonkids.org.au (D.R.L.); erika.sutanto@telethonkids.org.au (E.N.S.); kak-ming.ling@telethonkids.org.au (K.-M.L.); stephen.stick@health.wa.gov.au (S.M.S.); anthony.kicic@telethonkids.org.au (A.K.); 2Faculty of Health and Medical Sciences, University of Western Australia, Crawley 6009, Australia; leilanicullenn@gmail.com (L.C.); christopher.peacock@uwa.edu.au (C.S.P.); 3PathWest Laboratory Medicine WA, Fiona Stanley Hospital, Murdoch 6150, Australia; dianne.gardam@health.wa.gov.au; 4Department of Respiratory and Sleep Medicine, Perth Children’s Hospital, Nedlands 6009, Australia; 5Occupation and Environment, School of Public Health, Curtin University, Bentley 6102, Australia

**Keywords:** *Aspergillus*, inflammation, epithelium, cystic fibrosis, host response

## Abstract

*Aspergillus* is increasingly associated with lung inflammation and mucus plugging in early cystic fibrosis (CF) disease during which conidia burden is low and strains appear to be highly diverse. It is unknown whether clinical *Aspergillus* strains vary in their capacity to induce epithelial inflammation and mucus production. We tested the hypothesis that individual colonising strains of *Aspergillus fumigatus* would induce different responses. Ten paediatric CF *Aspergillus* isolates were compared along with two systemically invasive clinical isolates and an ATCC reference strain. Isolates were first characterised by ITS gene sequencing and screened for antifungal susceptibility. Three clusters (A−C) of *Aspergillus* isolates were identified by ITS. Antifungal susceptibility was variable, particularly for itraconazole. Submerged CF and non-CF monolayers as well as differentiated primary airway epithelial cell cultures were incubated with conidia for 24 h to allow germination. None of the clinical isolates were found to significantly differ from one another in either IL-6 or IL-8 release or gene expression of secretory mucins. Clinical *Aspergillus* isolates appear to be largely homogenous in their mucostimulatory and immunostimulatory capacities and, therefore, only the antifungal resistance characteristics are likely to be clinically important.

## 1. Introduction

*Aspergillus* can be the most frequently isolated pathogen in some young cystic fibrosis (CF) cohorts [1,2]. *Aspergillus* is highly efficient at releasing airborne conidia ubiquitously into the environment, making daily inhalation of conidia into the respiratory tract unavoidable [3]. The actual rates of *Aspergillus* infection are hard to quantify and likely underestimated, as clinical testing is not always routine and culture diagnostic methods are both insensitive and not well standardised [4,5]. Studies increasingly suggest Aspergillus may have an even earlier role in CF disease as a significant independent contributor to progression of structural lung disease, with evidence that infection can induce inflammation and mucus plugging in CF airways [6,7,8].

Among the *Aspergillus* species, *A. fumigatus* predominates in opportunistic CF fungal infections [9]. In the airways, *A. fumigatus* adapts physiologically and genetically to develop thermotolerance, makes metabolic adjustments to nutrient sources, and in some cases has resistance against antifungal therapy [3,10]. Unlike *P. aeruginosa* and other bacteria which can develop under selection pressures into transmissible clinical strains that disseminate across CF care centres [11], clinical strains of *Aspergillus* and other filamentous fungi appear to remain highly diverse [12] and genetic heterogeneity often features even within highly localised Aspergillosis outbreaks in hospital settings [13,14]. This may be because *Aspergillus* generate hyphae rather than conidia when colonising the lungs [15], reducing the means of efficient patient transmission. Spontaneous mutations that arise during asexual reproduction of *A. fumigatus* are another key source of genetic diversity in environmental isolates [16]. Therefore, several mechanisms drive genetic and biochemical heterogeneity in human *Aspergillus* clinical isolates.

The clinical observations linking *Aspergillus* with early inflammatory and structural disease pose the question as to whether early *Aspergillus* infections should be actively eradicated through therapy, which has been a long-standing question in older CF cohorts [17]. The major concerns to broad implementation of antifungal therapy in early CF are the rising incidence of azole resistance in environmental and clinical *Aspergillus* [4,10] and the contra-indication of some antifungals with CFTR (cystic fibrosis transmembrane conductance regulator) modulators [18], now an essential part of CF therapy [19]. Given the heterogeneity of *A. fumigatus* strains and the ability of this organism to adapt to host conditions, it would be beneficial to understand how *A. fumigatus* strains can vary in their ability to induce cell toxicity, inflammation, or mucus production. This is difficult to ascertain in vivo due to the issues surrounding the quantification and classification of fungal isolates and the low sampling frequencies in a rare disease such as CF.

Here, in vitro assessment of the responses by airway epithelial cells to *A. fumigatus* is informative as these cells form both the primary airway barrier and play an important innate immune role to alert macrophages and neutrophils to invading pathogens [20]. Several prior studies based upon in vitro investigations into the biology of the epithelial response to *A. fumigatus* have measured interleukin 6 (IL-6) and 8 (IL-8) concentrations and MUC5AC/B mRNAs in order to determine the induction of inflammatory signalling molecules and secretory mucins involved in mucous plugging [21,22,23,24,25], respectively. However, these studies did not compare many clinical isolates, typically employed high multiplicities of infection (MOI) that assess stimulation by conidia rather than germination, and utilised cell lines rather than primary airway epithelial cells (pAEC). To address this, we tested the hypothesis that different clinical strains of *A. fumigatus* would induce different inflammatory and mucin responses. We aimed to characterise multiple clinical *A. fumigatus* strains for their genetic diversity and antibiotic susceptibility profiles before evaluating the use of pAEC as a laboratory model of initial *Aspergillus* infection to screen the immunostimulatory and mucostimulatory capacity of clinical isolates in comparison to both one another and to laboratory reference *Aspergillus* strains.

## 2. Results

### 2.1. Isolate Clustering and Antifungal Susceptibility

All 12 clinical *Aspergillus* isolates (CFA001−010 and Inv1−2) were confirmed as *A. fumigatus* by colonial, microscopic morphology, and Matrix-Assisted Laser Desorption/Ionization-Time of Flight (MALDI -TOF) mass spectrometry (MS) (Bruker Version 3.1; Appendix A). Subsequent internal transcribed spacer (ITS) sequencing and alignment to GenBank identified the isolates as belonging to one of the following three recognised sequences: *A. fumigatus* S10 (Cluster A), *Aspergillus* spp. isolate CK392 (Cluster B), and *A. fumigatus* strain I (Cluster C; Appendix A). The *A. fumigatus* S10 strain was the most prevalently recognised sequence, featuring many of the clinical isolates (six CF isolates and both invasive isolates) and the ATCC (#46645) isolate. There was minimal evolutionary distance between the three identified *A. fumigatus* clades (Appendix A). Antifungal susceptibility data are presented in Table 1 with isolates grouped by their sequence-based clustering. Variability was evident for the azoles, with MICs ranging between 0.015 and 0.25 μg/mL for posaconazole and <0.015–0.5 μg/mL for itraconazole.

### 2.2. Monolayer Aspergillus Infections Show no Differences between Responses to Individual Isolates

Monolayer cultures of pAEC were established from bronchial brushings of three children with CF and three non-CF peers (Table 2). Notably, two of the three children with CF had positively cultured *Aspergillus* on prior BAL. Data from the monolayer stimulations were analysed by repeated measures non-parametric Friedman tests with multiple comparisons to the uninfected controls only. Overall, the inflammatory and cytotoxic responses induced after 24 h did not differ significantly according to individual *A. fumigatus* strains (Figure 1). No *A. fumigatus* isolate or ITS cluster was observed to induce significant IL-8 expression over control at either multiplicity of infection (MOI) of 0.01 or 1 (Figure 1A,B). For the cytotoxicity assessment, we reported lactate dehydrogenase (LDH) release in response to strains as a ratio to LDH release by the uninfected control. There was no interaction observed between cells and *A. fumigatus* isolates, with post-hoc testing of LDH ratios for individual isolates against the ATCC reference strain indicating that none were significantly different at inducing epithelial cytotoxicity. In addition, no significant differences were observed in IL-8 and LDH release across both MOIs. However, comparison between the non-CF and CF cohorts by the Mann–Whitney test found a significantly higher LDH release in the non-CF cohort (Figure 1C,D) at both MOI 0.01 (median ratio 1.31 vs. 1.15, *p* = 0.0164) and MOI 1 (median ratio 1.47 vs. 1.17, *p* < 0.0001).

### 2.3. Air–Liquid Interface (ALI) Infections also Showed no Inflammatory Differences between Isolates

Considering the similar inflammatory response to the different fungal isolates, only five strains were used for the more resource intensive air–liquid interface (ALI) exposure experiments. The four strains were selected in order to compare between CF isolates from different ITS clusters, an invasive isolate, a reference strain ATCC#46645 (ITS Cluster A), CF clinical isolate CFA002 (ITS Cluster A), CF clinical isolate CFA007 (ITS Cluster B), and one invasive aspergillosis isolate Inv1 (ITS Cluster A). Differentiated pAEC cultures were established from five non-CF and six CF donors (Table 3) by ALI culture for at least 28 days and then challenged with *A. fumigatus* at MOI 0.01. A lower dose was tested to model a typical in vivo daily dose of a few hundred inhaled conidia [3]. All isolates showed hyphal growth after 24 h featuring similar morphology when viewed by brightfield microscope (Figure 2).

The media from the basal compartment of ALI cultures were found to be significantly positive for *Aspergillus* galactomannan after 24 h challenge by *A. fumigatus* isolates (all *p* < 0.05; Figure 3A). Galactomannan levels were similar between the non-CF and CF cohorts. The 24 h basal media samples were then assessed for the release of inflammatory markers IL-8 and IL-6 from the epithelium (Figure 3B and Figure 3C, respectively). Data from the stimulations of differential cultures were also analysed by repeated measures of the non-parametric Friedman test with multiple comparisons between all conditions. We observed that CFA007 significantly increased IL-8 expression (Figure 3B) above uninfected control levels in both non-CF (mean 29,639 pg/mL ±12,470 vs. 8537pg/mL ±1510; *p* < 0.05) and CF cultures (mean 37,599 pg/mL ±36,412 vs. 7229 pg/mL ±6763; *p* < 0.05). However, there were no significant differences observed between the four isolates. Furthermore, no individual isolate was different from the uninfected control or to other isolates in their stimulation of epithelial IL-6 production (Figure 3C). We noted with interest that, amongst the individual epithelial cultures of the CF cohort, there was a bimodal distribution for both IL-8 and IL-6 production (Figure 3B,C). Inspection of the donor characteristic revealed that the two cultures with strong cytokine responses to *Aspergillus* were both from *Aspergillus* naïve CF donors aged less than 6 months of age. Of the four remaining cultures, which did not generate an inflammatory response, two were from individuals with a history of *Aspergillus* culture positivity (shown previously in Table 3).

### 2.4. Aspergillus Isolates Did Not Induce Gene Expression of Secretory Mucins

Due to the association of *Aspergillus* in early CF with mucus plugging on lung CT, this study next investigated the ability of the *A. fumigatus* isolates to induce mucin production in the differentiated cell cultures. Consistent with the similar inflammatory responses between isolates, the expression of both MUC5AC and MUC5B genes was similar and no isolate significantly induced expression of either gene (Figure 4A and Figure 4B, respectively).

## 3. Discussion

This study investigated whether there are significant differences in the immunostimulatory or mucostimulatory capacity between individual clinical isolates of *A. fumigatus*. We characterised 10 paediatric CF clinical isolates and two invasive *Aspergillus fumigatus* strains by antibiotic susceptibility profiles and genetic diversity before evaluating inflammatory and viability responses by pAEC cultures. The antifungal susceptibility profile was diverse across the different clinical isolates. Within our 10 clinical CF isolates, we identified three distinct genetic clusters from ITS analysis. The genetic and susceptibility diversity in our CF clinical strains of *A. fumigatus* is consistent with a previous larger study [12]. However, our results indicated that this diversity was not associated with different capacities in inducing inflammatory or mucin responses from primary epithelium. This study tests a panel of clinical *A. fumigatus* isolates from early CF disease on numerous pAEC cultures in both submerged monolayer and differentiated ALI modalities, permitting effective comparison to the prior literature [21,22,23,24,25].

The clinical isolates did not vary in their immunostimulatory capacity. In our submerged epithelial model, we assessed the 24 h inflammatory response to *A. fumigatus* germination at MOI 0.01 or 1. Equivalent to 300 and 3 × 10^4^ conidia, respectively, we selected these MOIs as Reece and colleagues demonstrated cytotoxicity is induced at MOI 2 or greater [24], and Balloy and colleagues showed that germination was the main inducer of the IL-8 response [21]. Similar to prior studies [21,22], we also carried out our study on three independent experiments in triplicate. The key point of difference is that we utilised pAEC from young children rather than cell lines. Our experimental model resulted in conidial germination and no cytotoxicity, but significant increases in IL-8 production were not observed. This included measurable baseline production in both cohorts that did not significantly differ and is similar to previous observations using primary airway cell models. [27,28,29]. The methodology varies between the studies that have reported an IL-8 response to *A. fumigatus* conidia by submerged airway epithelium [21,22,24]. Reihill and colleagues assessed MOIs 1 to 250 and observed that only CFBE cells generated an IL-8 response if the MOI was 25 or greater [22]. This contrasts with Balloy et al. who used an MOI 6 with non-CF BEAS-2B cells [21] and Reece et al. who reported IL-8 release by CFBE cells in response to MOI 2 [24]. We interpret our finding that at MOI 1, none of the 10 clinical CF isolates which represented three different genetic lineages were able to consistently induce an inflammatory response in either CF or non-CF epithelium, as an indication that *A. fumigatus* isolates are largely homogenous in their immunostimulatory capacity.

When screening a subset of our isolates in the differentiated-ALI culture model, we again observed no significant differences between isolates in IL-8 or IL-6 secretion. While there are several publications featuring infection of differentiated cultures with *Aspergillus* [23,25,30,31,32], to our knowledge we are the first study to measure inflammatory responses. We showed through brightfield imaging of intact inserts and galactomannan measurements in basal media that effective germination had occurred. An unexpected finding in the differentiated model was that the CF epithelial response to *A. fumigatus* appeared to be consistently different in pAEC cultures established from the pre-school aged CF donors, with only pAEC from the younger 3 month old participants generating cytokine levels similar to non-CF cultures. We were not sufficiently statistically powered to validate this, but it hints at epigenetic changes in CF pAEC with age and/or infection that result in altered inflammatory pathway signalling to *A. fumigatus*. The transcriptomic response by primary ALI cultures to *Aspergillus* was recently described [25], but this study did not feature CF epithelium. With *Aspergillus* being a lifelong but poorly understood aspect of CF lung disease [33] and the recognition that genetic and epigenetic factors can modify CF disease (reviewed in [34]), there is a strong rationale to further investigate our observation with a larger number of donors for appropriate statistical power and the molecular techniques required.

The use of differentiated pAEC cultures was also performed in our study to facilitate investigation into mucin production by airway epithelium challenged by *A. fumigatus*. While *Aspergillus* is associated with early CF mucous plugging in vivo [6,35], in our model none of the *A. fumigatus* isolates induced either mucin gene expression or obvious protein changes in stained culture sections after 24 h incubation. To our knowledge, we are the first to assess mucin responses to live *Aspergillus*, as prior studies have utilised extracts or components of *A. fumigatus* [23,36]. We elected to measure the response at 24 h as Oguma et al. reported MUC5AC expression was maximally increased by extracts at this timepoint [23]. The later study by Kim et al. on the effect of *A. fumigatus* β-glucan on mucin gene expression chose an earlier assessment of 8 h, finding that only MUC5B expression and not MUC5AC was increased [36]. It is possible that, in our model, the isolates did not produce sufficient factors to initiate the mucin response and/or that the endpoint was too early. It has been demonstrated in a rodent model that chronic *A. fumigatus* infection for 7 days or longer results in goblet cell hyperplasia and increased mucin content [37]. However, there are multiple practical challenges surrounding epithelial cytotoxicity and mucous collection that will need to be addressed before conducting long-term in vitro infections. Instead, the quantification of the *A. fumigatus* proteases, β-glucan, and other putative factors in human clinical samples and appropriate CF animal models must also be performed in order to improve our understanding of which mechanisms underlie *Aspergillus* associated mucous plugging and their treatability.

In conclusion, the central message from our study would be the critical use of relevant *A. fumigatus* strains in laboratory models. The use of common reference strains should be sufficient when characterising early CF mucosal responses to *Aspergillus*, unless the research question specifically relates to antifungal resistance, which may require greater diversity among isolates. It is possible that strains may vary in other aspects, for example immunological escape from phagocytes [38], but this is beyond the scope of our epithelial focused study and there is no literature in this regard. Additionally, *A. fumigatus* strains isolated from adults with CF may display more variable immunostimmulatory capacity as a consequence of prolonged environmental adaptation and interactions with other CF microbes [39]. The summary for clinicians is that the presence of *Aspergillus* is more important to early CF lung pathology than the strain of *Aspergillus* and that typing should remain orientated to antifungal testing, ideally with a broader focus than the current azole testing.

## 4. Materials and Methods

### 4.1. Cell Culture, Media, and Reagents

The pAEC were obtained from the tracheal brushings of CF participants undergoing their annual clinical assessment through the Australian Respiratory Early Surveillance Team for Cystic Fibrosis (AREST CF) program (Perth Children’s Hospital, Nedlands, Western Australia). Non-CF pAECs were obtained from healthy participants undergoing non-respiratory-related surgery through the Western Australian Epithelial Research Program (WAERP; St John of God Hospital, Subiaco, Western Australia). Prior to experiments, cryopreserved stocks of pAEC were thawed and then expanded at 37 °C/5%CO_2_ in T75 flasks pre-coated with fibronectin/collagen and irradiated NIH-3T3 fibroblasts, with initial seeding of 5000 cells per cm^2^, as previously described [40]. From these flasks, cells were detached by trypsinization and, for submerged experiments, cultured to 70% confluence in 96-well plates in Bronchial Epithelial Cell Growth Medium (BEGM) (Lonza) prior to infection. For differentiated pAEC cultures, cells were seeded onto collagen-coated 12-well 0.4 μm pore inserts and maintained for 28 days at the air–liquid interface (ALI) in PneumaCult ALI media (STEMCELL Technologies, Vancouver, BC, Canada). Prior to performing all *Aspergillus* infections, epithelial cultures were acclimatised for 24 h to media without antibiotics, hydrocortisone, and epithelial growth factors, as these can dampen immune responses [41,42].

### 4.2. Aspergillus Isolates

In total, 13 *Aspergillus fumigatus* isolates were used in this study. Ten isolates were obtained from PathWest Laboratory Medicine WA (Nedlands, Western Australia) from paediatric CF respiratory samples that were culture positive for *Aspergillus fumigatus.* Two were historical non-CF isolates obtained from invasive aspergillosis cases at Fiona Stanley Hospital (Perth), from the vitreous fluid of an abattoir worker and from the right hip nodule of a heart transplant patient who presented with multiple tender erythematous nodules. The ethical use of these cultures was classified as ‘negligible risk research’ in accordance to Section 2.3 of the 2015 *Australian National Statement on Ethical Conduct*. The *A. fumigatus* reference strain ATCC #46645 was sourced from American Type Culture Collection (ATCC, Manassas, VA, USA). Isolates were propagated via culturing on slopes of Sabouraud Dextrose Agar with Chloramphenicol (SAB + C) free from bacterial contamination.

### 4.3. Antifungal Susceptibility

All isolates were screened for their antifungal susceptibility by using the Sensititre YeastOne plate inoculation procedure (Thermo Fisher Scientific, Melbourne, VIC, Australia, #Y010). For this assay, conidia solutions were made in 0.05% saline instead of Phosphate Buffered Saline (PBS), and the turbidity of the conidia solution was adjusted with the addition of saline to achieve 80–82% transmittance measured at OD530 (equivalent to inoculum 0.6–5 × 10^6^ CFU/mL). A 100 μL aliquot of this solution was transferred into 11 mL of Sensititre YeastOne broth, then 100 μL of this mixture was aliquoted into each well. To confirm inoculums were correct, a sample was taken from the positive control well and streaked onto a SAB agar plate, upon which 50–500 fungal colonies were required to be present after 24 h incubation.

### 4.4. Isolate Typing

All isolates were typed for identification by using two methods: (1) MALDI-TOF and (2) internal transcribed spacer (ITS) gene sequencing. For MALDI-TOF identification, isolates were inoculated in SAB broth and incubated for 72 h at room temperature. The conidia pellet was then collected and dried before adding formic acid and acetonitrile. This solution was added to MALDI target plates in duplicate and overlaid with 1 μL alpha-cyano-4-hydroxycinnamic acid (HCCA), then analysed by using Bruker’s Microflex LT system. An Escherichia coli extract was used as a bacterial test standard (BTS) to ensure correct matching of protein peaks by the machine. An established ITS sequencing protocol was performed [43] whereby the conidia pellet of each isolate was inoculated in 0.5 units per litre (U/L) of lyticase, and nucleic acid was extracted using the MagNA Pure Total Nucleic Acid Extraction Kit (Roche Molecular Systems, Sydney, NSW, Australia, #03038505001). Prior to ITS sequencing, PCR products were diluted to an optimal concentration with the addition of nuclease free H2O. The ITS polymerase chain reaction (PCR) reaction mix included primers V9D (5′ TTA AgT CCC TgC CCT TTg TA 3′) and LS266 (5′ gCA TTC CCA AAC AAC TCg ACT C 3′). The reactions were conducted in 96 well plates on a Roche LightCycler 480 Real-Time PCR system, with the positive control (Candida albicans ATCC 14053C) optimised to approximately 22 cycles. The ITS PCR products were then sequenced and sequences searched against the GenBank database (NCBI, Bethesda, MD, USA) using the BLAST standard the nucleotide–nucleotide basic local alignment search tool [44].

### 4.5. Aspergillus Epithelial Infection Model

Conidia solutions for infection experiments were made by vigorously vortexing conidia collected from slopes in a 0.5% *v/v* Tween20/PBS solution, then resting the solution for five minutes for the sedimentation of hyphal fragments. The upper half of the solution was considered the hyphae/mycelia free portion and used for the laboratory experiment. Conidia concentrations were calculated by automated cell counter. The initial screening of airway epithelial responses to all isolates was achieved by stimulating submerged pAEC cultures (n = 3 non-CF and 3 CF) with conidia solutions at MOI 0.01 (0.3 × 10^3^) and 1 (3 × 10^4^) in quadruplicate. After initial stimulation at 37 °C/5%CO_2_ for 6 h to permit conidia swelling and adhesion [45], the media/conidia were removed to simulate mucociliary clearance and immune mediated clearance, and fresh media replaced for a further 18 h incubation. After 24 h from the initial conidia stimulation, culture media were collected, filtered, and archived at −80 °C. In follow-up experiments, these findings were validated by using differentiated airway epithelial cultures, achieved by culturing pAEC (n = 5 non-CF and 6 CF) at the air–liquid interface (ALI) using our prior methodologies [36,46], that better reflect the respiratory mucosa. For stimulation, the differentiated pAEC cultures were exposed apically to conidia at MOI 0.01 in 50 µL of PBS (1.5 × 10^3^), with uninfected controls receiving 50 µL PBS vehicle. All stimulations were performed on duplicate culture inserts. Again, cultures were initially incubated at 37 °C/5%CO_2_ for 6 h before the apical surface was washed twice with 100 µL PBS to remove unbound conidia. The basal media were also collected, and fresh media replaced it for an additional 18 h incubation at 37 °C/5%CO_2_. At 24 h post conidia exposure, images of the apical surface for all inserts were acquired by brightfield microscopy (Nikon Ts2R Fluorescent Inverted Microscope). Basal media were collected, filtered (0.22 µm), and archived at −80 °C. One complete cell culture insert was fixed in 4% *v/v* formalin solution before wax embedding; the other insert was lysed with buffer RLT (QIAGEN, Melbourne, VIC, Australia, #79216) and stored at −80 °C.

### 4.6. ELISA Analysis

Epithelial cell toxicity was determined by measuring lactate dehydrogenase (LDH) release using the CytoTox 96 assay (Promega, Sydney, NSW, Australia, #G1780). The neutrophil chemotactic cytokine interleukin-8 (IL-8) was measured using the OptEIA Human IL-8 ELISA set (BD, #555244) according to manufacturer’s instructions, and IL-6 was measured using an in-house time-resolved fluorometry detection system [47]. *Aspergillus* galactomannan was examined by the Platelia *Aspergillus* Ag sandwich microplate assay (Bio-Rad, Sydney, NSW, Australia, #62794) with positivity determined by a positive index of >1.5.

### 4.7. Histology and qPCR

For all ALI exposures, one of the two inserts were formalin fixed (4%) and wax embedded for histological investigation via Grocott’s silver stain and Periodic Acid-Schiff stain [48]. RNA was collected from the remaining insert and used to investigate MUC5AC and MUC5B expression by qPCR. RNA was extracted using PureLink^®^ RNA (Thermo Fisher Scientific, Melbourne, VIC, Australia) as per manufacturer’s protocol with the addition of DNase treatment to remove potential genomic DNA contamination. Total RNA was then eluted with 40 μL RNase free water and RNA purity, and yield was determined by using a NanoDrop. Gene expression was analysed by two-step reverse transcriptase–polymerase chain (RT-PCR) reactions. cDNA was synthesised by using random hexamers and Multiscribe Reverse Transcriptase (Thermo Fisher Scientific, Melbourne, VIC, Australia) under the following condition: 25 °C for 10 min, 48 °C for 60 min, and 95 °C for 5 min. The expressions of target and housekeeping genes were then assessed via real time qPCR, performed using a Quanstudio™ 7 Flex with the following parameters: 1 cycle of 60 °C for 2 min, followed by one cycle of 95 °C for 10 min to activate polymerase, and followed by 40 cycles of denaturation at 95 °C for 15 s and then annealing and elongation at 60 °C for 1 min. Predesigned Taqman primer/probe sets used for this study were MUC5A (Hs01365616 M1), MUC5B (Hs00861595 M1)), and cyclophilin A PPIA (Assay ID Hs99999904_m1). A 10 μL final reaction containing a 2.5 μL cDNA template, 5 μL Taqman™ Universal PCR Master Mix buffer, and 0.5 μL of Taqman™ primer/probes and RNase-free water was then prepared for each sample in duplicate. The expression of target genes was determined as 2(−ΔΔCT) values relative to the expression of housekeeping gene, PPIA.

### 4.8. Statistical Analysis

Data were tested for normality via a Shapiro–Wilk test and visualisation of residual and QQ plots. Raw data were analysed by appropriate parametric or non-parametric tests with repeated measures matched by the epithelial donor. The results for each isolate were only compared to uninfected control (or ATCC #46645 for LDH release), and multiple comparisons were corrected by using Dunn’s post-hoc tests. All analyses were performed using GraphPad Prism version 9.01.

## 5. Conclusions

Clinical isolates of *A. fumigatus* do not vary in their ability to stimulate IL-8 and IL-6 or MUC5AC and MUC5B. The presence of *Aspergillus* is more important to CF lung pathology than the strain of *Aspergillus*, and that typing should remain orientated to antifungal testing, ideally with a broader focus than the current azole testing. Laboratory-based experiments into understanding the host biology during *A. fumigatus* infections should aim for use of common reference strain to aid reproducibility.

## Figures and Tables

**Figure 1 pathogens-10-01020-f001:**
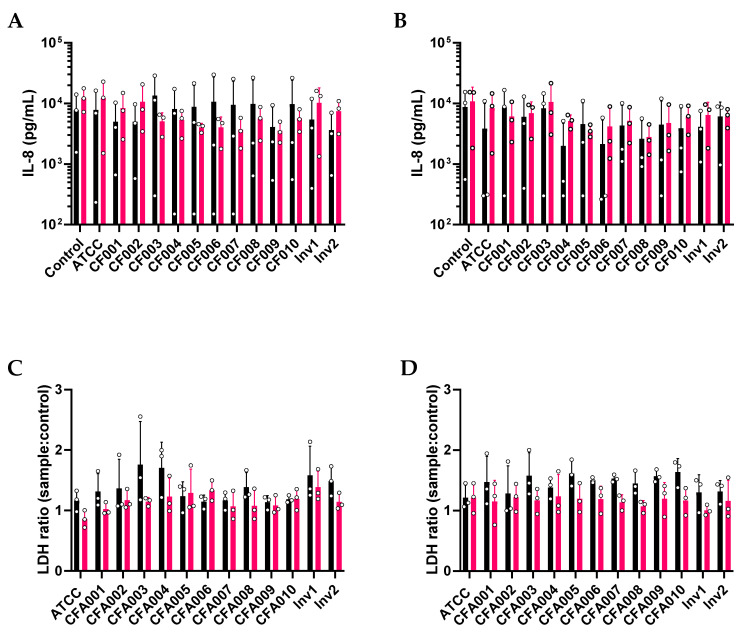
*Aspergillus* failed to induce an inflammatory response by primary airway epithelial cells. No individual isolate or cluster induced the release of the inflammatory cytokine IL-8 when compared to control, either at MOI 0.01 (**A**) or 1 (**B**) after 24 h of exposure. Similarly, no individual isolate consistently induced significant lactate dehydrogenase (LDH) release at MOI 0.01 (**C**) and 1 (**D**). However, at a cohort level, non-CF epithelium were observed to release significantly greater LDH than CF epithelium at MOI 0.01 (**C**; *p* = 0.0164) and MOI 1 (**D**; *p* < 0.0001). Note: Non-CF results are indicated with black bars and CF with red bars. Data points indicate results from individual pAEC cultures.

**Figure 2 pathogens-10-01020-f002:**
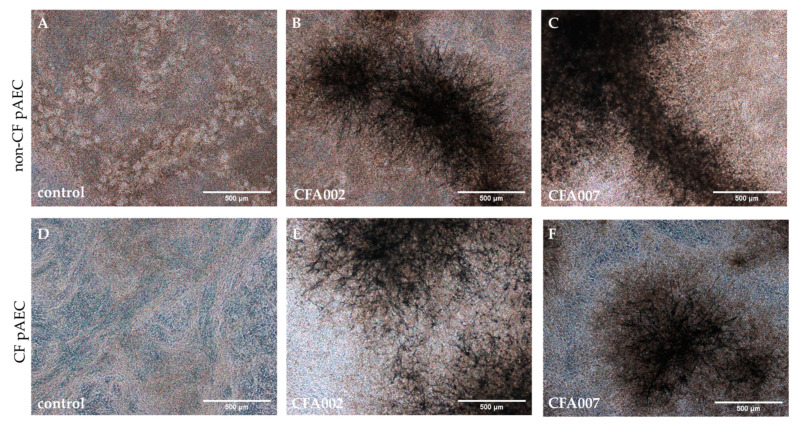
*Aspergillus fumigatus* achieved hyphal growth on non-CF and CF epithelium. After 24 h incubation, the apical surface of non-CF (**A**–**C**) and CF (**D**–**F**) differentiated cultures were live imaged by brightfield microscopy to record localisation across the apical surface. Representative images are shown of uninfected control inserts (**A**,**D**) and CF *A. fumigatus* clinical isolates CFA002 (**B**,**E**) and CFA007 (**C**,**F**).

**Figure 3 pathogens-10-01020-f003:**
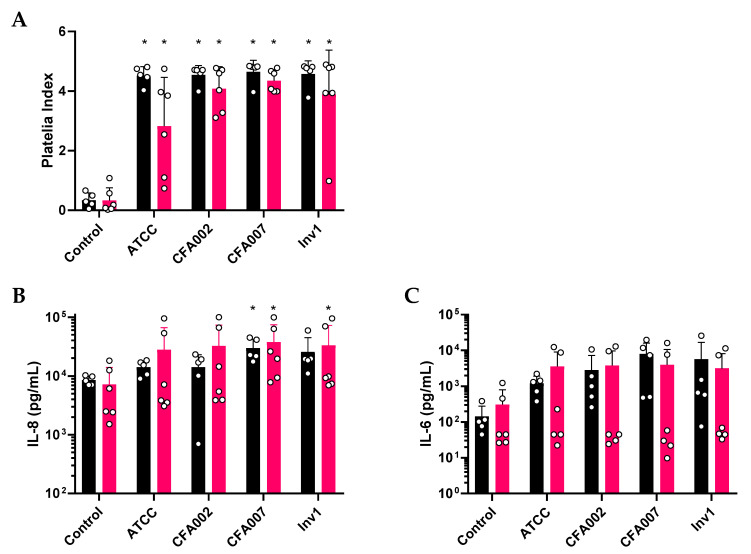
Different *A. fumigatus* strains induce similar inflammatory responses in differentiated primary epithelium. After non-CF and CF pAECs were cultured with *Aspergillus* for 24 h and the levels of the *A. fumigatus* exoantigen galactomannan were found to be significantly increased for all *Aspergillus* isolates (**A**). Concentrations of IL-8 in the basal media did not significantly differ between the four isolates assessed, with only the CF paediatric *A. fumigatus* isolate CFA007 significantly increased compared to uninfected controls in both non-CF and CF cultures (**B**). No difference in IL-6 release compared to uninfected controls was observed for any isolate in either cohort (**C**). Note: Non-CF results are indicated with black bars and CF with red bars. Data points indicate results from individual pAEC cultures. * represents a *p* value < 0.05.

**Figure 4 pathogens-10-01020-f004:**
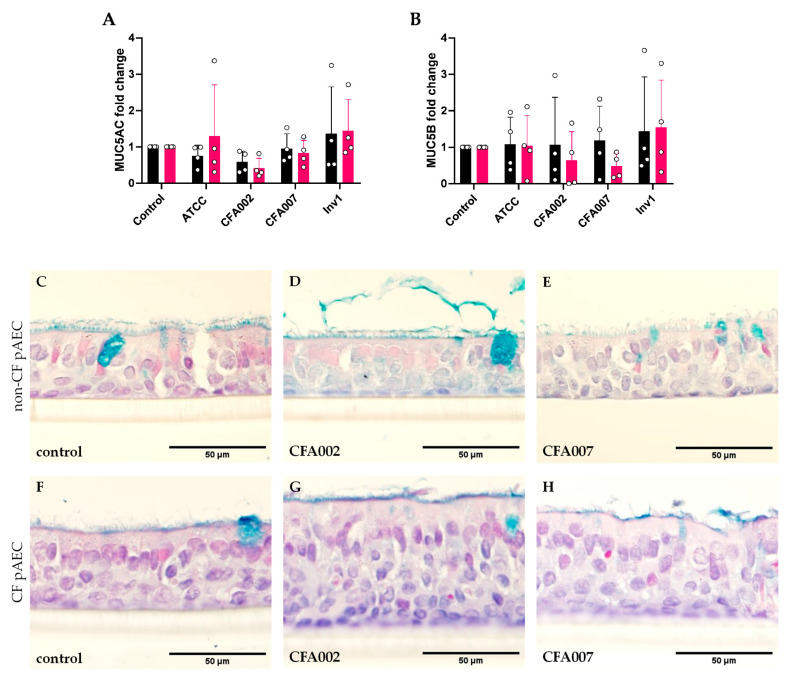
*Aspergillus* isolates at MOI 0.01 did not induce mucin gene or protein expression. Following 24 h culture with *A. fumigatus* isolates, epithelial RNA was analysed for *MUC5AC* (**A**) and *MUC5B* (**B**) gene expression by qPCR. Representative images of Alcian Blue stained sections (**C–H**) demonstrating no significant changes in mucin protein between uninfected control epithelium (**C**,**F**) and epithelium challenged by *Aspergillus* CFA002 (**D**,**G**) and CFA007 sections (**E**,**H**). Non-CF sections on the top row (**C**–**E**) and CF sections on the bottom (**F**–**H**). Note: Non-CF results are indicated with black bars and CF with red bars. Data points indicate results from individual pAEC cultures.

**Table 1 pathogens-10-01020-t001:** ITS Clustering and Antifungal Susceptibility.

ITS Cluster	Isolate ID	MIC Endpoint (µg/mL)		
		PZ *ND*	VOR *1*	IZ *1*
A—*A. fumigatus* S10	CFA 001	0.12	0.25	0.25
	CFA 002	0.03	0.25	0.06
	CFA 004	0.03	0.25	0.06
	CFA 005	0.015	0.25	<0.015
	CFA 006	0.015	0.25	0.03
	CFA 010	0.015	0.25	0.03
	Inv 1	0.25	0.5	0.5
	Inv 2	0.12	0.5	0.25
B—*Aspergillus* spp. isolate CK392	CFA 007	0.03	0.5	0.06
	CFA 008	0.015	0.25	0.12
	CFA 009	0.015	0.12	<0.015
C—*A. fumigatus* strain I	CFA 003	0.015	0.25	0.03

The minimum inhibitory concentrations (MIC) are presented in µg/mL for each antifungal (PZ = posaconazole, 0.008–8 µg/mL; VOR = voriconazole, 0.008–8 µg/mL; IZ = itraconazole, 0.015–16 µg/mL). The epidemiological cut-off (ECOFF) value for each antifungal [26] is in italics below the abbreviation. ND = not determined; IR = inherently resistant. Isolates of *Aspergillus fumigatus* are listed by row. CFA = paediatric CF clinical isolates; Inv = systemically invasive isolates from aspergillosis cases.

**Table 2 pathogens-10-01020-t002:** Participant Demographics for Submerged Monolayer Culture Experiments.

	Non-CF	CF
Number (n)	3	3
Age (years) (mean ± SD)	2.8 ± 0.7	2.8 ± 2.9
Gender	1/3 male	2/3 male
Current *Aspergillus* BAL culture	Not tested	0/3
Previous *Aspergillus* BAL culture	Not tested	2/3 (*A. fumigatus*; *A. niger*)

**Table 3 pathogens-10-01020-t003:** Participant Demographics for Differentiated ALI Culture Experiments.

	Non-CF	CF
Number (n)	5	6
Age (years) (mean ± SD)	2.9 ± 0.8	2.8 ± 2.5
Gender	3/5 male	1/6 male
Current *Aspergillus* BAL culture	Not tested	0/6
Previous *Aspergillus* BAL culture	Not tested	2/6 (*A. fumigatus*; *A. niger)*

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
