# Peer review of "Cystic Fibrosis Clinical Isolates of Aspergillus fumigatus Induce Similar Muco-inflammatory Responses in Primary Airway Epithelial Cells"

_pathogens, 2021, doi:10.3390/pathogens10081020_

Round 1

Reviewer 1 Report

The manuscript has substantially improved.

Minor comments:

Legend fig 1 and fig 3 need to be amended as graphs have changed from lines to bars; explain the red and black colour of the bars.

Author Response

C1: The manuscript has substantially improved.
R1: We thank Reviewer 1 for their review and feedback.
Minor comments:
C2: Legend fig 1 and fig 3 need to be amended as graphs have changed from lines to bars; explain the red and black colour of the bars.
R2: Legends on figure 1, 3 and 4 have been corrected to clearly explain the new graph format. Bar colours now refer to CF and non-CF data, with results from individual pAEC cultures indicated by small circles.

Reviewer 2 Report

A major issue with the paper are the strains used and which were isolated from CF patients. The authors present the strains as being the isolates that indeed do grown in the CF lung and cause of problems in CF lung. However, isolating A fumigatus from lung sputum or BAL represents strains that either ended up there just before sampling after inhalation of conidia as well as isolates that actually do develop there and which could even adapt in time. The authors do not separate these isolates as such and therefor the CF isolates could be general representative of conidia present in the air a well. One must be particular careful in interpreting these data since one can seriously doubt whether these isolates are true representatives of colonizers of CF lung. The paper is comparing a set of isolates which in itself is interesting, but conclusions as if these isolates indeed represent colonizers needs further investigations. The authors should adapt the manuscript in light of this remark.

Line 114 data to data.. adapt sentence

Fig 1 and Fig 3 why is the control IL-8 response so high? The authors should make a remark on this. Are these cells somehow exposed to IL-8 inducing agents in the medium?

Fig 1 some isolates from CF patients at MOI 1 show reduced IL-8 induction (CF04/06/08) and seem close to significantly different, this seems rather remarkable. If you compare IL-8 induction between MOI 0.1 and 1 it is also remarkable that at MOI 1 a subset of the CF isolates induce in general lower amounts of IL-8 as compared to MOI 0.01, this seems strange as higher amounts of conidia would in general would result in increased induction. Did the authors compare these two MOI sets for significant differences? The control is stable in both experiments.  It looks as if some CF isolates are actually reducing IL-8 induction at higher MOI. The LDH values do not differ. The authors do not comment on this observation

Considering the above, it seems strains to choose only 2 strains for ALI experiments as depicted in Fig 3 were actually also an MOI of 0.01 was used. Such experiments should have been performed with some additional isolates and also at MOI 1 for comparison.

Fig 4 panel A and B show overlapping images, a line and bar graph runs through each other. Also in this case MOI 1 and some other CF isolates should have been compared. The set of strains is already very limiting to make such strong conclusion as stated in line 454. The conclusion described at line 479 are actually based on 2 strains from CF patients, this is an overinterpretation of the results.

Line 517-519 the data present in this paper encompass only a limited analysis in vitro of the many aspects and responses that can occur. The statement made in 517-519 that one should not include many different genetic lineages is to strong.

Author Response

C1: A major issue with the paper are the strains used and which were isolated from CF patients. The authors present the strains as being the isolates that indeed do grown in the CF lung and cause of problems in CF lung. However, isolating A fumigatus from lung sputum or BAL represents strains that either ended up there just before sampling after inhalation of conidia as well as isolates that actually do develop there and which could even adapt in time. The authors do not separate these isolates as such and therefor the CF isolates could be general representative of conidia present in the air a well. One must be particular careful in interpreting these data since one can seriously doubt whether these isolates are true representatives of colonizers of CF lung. The paper is comparing a set of isolates which in itself is interesting, but conclusions as if these isolates indeed represent colonizers needs further investigations. The authors should adapt the manuscript in light of this remark.
R1: We thank Reviewer 2 for their review and feedback. We have now amended portions of our concluding statements to clarify that our experiments provide insights into A. fumigatus infection in early CF (lines 311, 313, 317, and 321). It is true that these childhood strains often originate from the environment, but they are still increasingly linked to early lung disease. These early strains may indeed not be as pathogenic as strains that have colonised adults, which have had longer time to adapt to the CF lung environment.
C2: Line 114 data to data.. adapt sentence
R2: The sentence in question (now line 42) has been edited for clarity.
C3: Fig 1 and Fig 3 why is the control IL-8 response so high? The authors should make a remark on this. Are these cells somehow exposed to IL-8 inducing agents in the medium?
R3: Laboratory airway cell cultures have been documented to constitutively release IL-8 (Marshall et al 2001, J Immunol). In the literature, studies using airway cells often detect baseline levels of IL-8 in uninfected controls, and our baseline IL-8 concentration falls within a range similar to baseline levels reported in several citations in this manuscript (103-104 pg/mL). This is now stated in Line 262.
C4: Fig 1 some isolates from CF patients at MOI 1 show reduced IL-8 induction (CF04/06/08) and seem close to significantly different, this seems rather remarkable. If you compare IL-8 induction between MOI 0.1 and 1 it is also remarkable that at MOI 1 a subset of the CF isolates induce in general lower amounts of IL-8 as compared to MOI 0.01, this seems strange as higher amounts of conidia would in general would result in increased induction. Did the authors compare these two MOI sets for significant differences? The control is stable in both experiments. It looks as if some CF isolates are actually reducing IL-8 induction at higher MOI. The LDH values do not differ. The authors do not comment on this observation.
R4: We performed multiple t-tests with a repeat comparisons correction and found no significant difference in IL-8 or LDH between MOIs in Figure 1 (now stated in line 116). In addition, we have now included the individual data points on the bar graphs to show the spread of the data, which better illustrates why the conditions are not significantly different. With the spread of data in these experiments it becomes difficult to conclude if IL-8 is being truly reduced.
C5: Considering the above, it seems strains to choose only 2 strains for ALI experiments as depicted in Fig 3 were actually also an MOI of 0.01 was used. Such experiments should have been performed with some additional isolates and also at MOI 1 for comparison.
R5: Our ALI experiments were limited by the high cost and resources required for large screens. As such two CF isolates, an invasive isolate, and a laboratory strain were used at a single MOI, to answer the simple question if host tissue responses would be any more or less variable in differentiated tissue versus undifferentiated cultures. We chose CF isolates from two separate ITS clusters, while a lower MOI was selected to model doses of conidia from daily inhalation (now stated in lines 172 and 177).
C6: Fig 4 panel A and B show overlapping images, a line and bar graph runs through each other. Also in this case MOI 1 and some other CF isolates should have been compared. The set of strains is already very limiting to make such strong conclusion as stated in line 454. The conclusion described at line 479 are actually based on 2 strains from CF patients, this is an overinterpretation of the results.
R6: We have checked the latest version of the manuscript document and found no issue upon conversion to PDF, it should no longer have overlapping graphs. Reiterating our response to the previous comment, cost and resource intensiveness were limiting factors in our screen by ALI cultures and isolates and MOIs were selected according to the previously mentioned criteria. We have edited the statements in question (now lines 249 and 273, respectively) to better reflect the implications of this study.
C7: Line 517-519 the data present in this paper encompass only a limited analysis in vitro of the many aspects and responses that can occur. The statement made in 517-519 that one should not include many different genetic lineages is to strong.
R7: We have rephrased the statement to correctly summarise our key message (now lines 311-315).
We hope you will find the amendments appropriate and that the manuscript is now suitable for publication in Pathogens. We look forward to hearing from you in due course.
With kind regards.

This manuscript is a resubmission of an earlier submission. The following is a list of the peer review reports and author responses from that submission.

Round 1

Reviewer 1 Report

In the manuscript entitled “Cystic Fibrosis Clinical Isolates of Aspergillus fumigatus Induce Similar Muco-inflammatory Responses in Primary Airway Epithelial Cells”, the authors tested whether different clinical strains of A. fumigatus would induce different inflammatory and mucin responses in epithelial cells.

In opinion of this reviewer, the hypothesis is interesting, the manuscript is well explained, and some aspects of the scientific research are interesting, as the fact that it has been carried out with primary cultures directly isolated from patients instead of cell lines.

However, there are some major problems that make that conclusions obtained could not be totally probed.

It is very concerning, for example, the number of samples used. The authors used 3 CF and 3 non-CF samples. The results show high variability and in some cases 2 out of the 3 samples show a tendency but as the third sample does not, differences are not statistically significant (this happens, for example with the CF003, among others, in more than one image). In these cases, a higher number of samples could solve the problem and make possible to observe more differences.

On the other hand, the results don’t have standard deviation. How many biological or/and technical replicates were used?. If I am not wrong, they only do two technical replicates and no biological one. I think that this is not enough, above all takin into consideration the low number of samples, to obtain conclusions. It is very important to show deviations to have a better idea of the representativeness of the data.

I see interesting that Inspection of the donor characteristic revealed that the two cultures with strong cytokine responses to Aspergillus were from Aspergillus naïve CF donors, and that of the four remaining cultures, which did not generate an inflammatory response, two were from individuals with a history of Aspergillus culture positivity. With more samples, this fact could be proved and widely discussed.

Minor comments:

Line 57: P. aeruginosa in italics

Line 97: A. fumigatus in italics

Reviewer 2 Report

The authors have described the absence of epithelial cells responses upon fungal challenge using CF and non CF A. fumigatus isolates. Even the results look interesting, there are some flaws in the experimental design that would required further clarification.

L57,L96: Italics in species names

L105: do the authors mean between the three A. fumigatus clades? A reference to support the grouping should be required.

L170: the authors state that there are not differences in fungal growth of the Aspergillus strains – epithelial cells ALI co-culture interactions but this has not been quantified and the pictures are very blurry.

L190: It is not clear whether the results from the first section have been done using submerged culture and then the model was refined to an ALI culture. In that case, what is the adventage of including the results of the submerged culture in the manuscript? Probably the authors can be just focused in the ALI model that it is the one that is far mor physiological relevant. Additionally, it is not clear what readouts did the authors use to guarantee that the monolayers in ALI culture were fully differentiated and confluent. This should be described and data should be shown to support that viability of the monolayers after differentiation. Additionally, the authors state that the basal media was positive for galactomannan, did they observed fungal penetration to the basal compartment? In that case, was that quantified? Were there any differences between CF and non-CF monolayer confluency after infection? Additionally it is not clear why there are not basal controls for the CF epithelial cells in Aspergillus challenge experiments. Additionally, it would look better if the results were grouped by disease and stain instead of the graphs with the lines that are a little bit messy and difficult to interpret and get any conclusion about significance. Similarly, it would be better if mucin production was determined by slotblot or westernblot so quantitative data can be shown.

L358: details on ethical approval are missing

L383: details about where the experiments were conducted are not necessary

L397: Could the authors confirm whether the ITS PCR products were sequenced using sanger sequencing and if the primers uses were taken from the literature or developed in house. In case they were designed in house and the PCR optimised, can the authors provide with details on sensitivity, specificifcity, target etc?

L404-L408: please revise academic writing style as many details are unnecessary for a publication.

L407: it is not clear for me the rationale behind removing unbound conidia after 6 h exposure. Why did the authors not spin down the plate at the beginning in order to synchronise exposure of all conidia to the epithealial cells. Additionally, do the authors know how many conidia were unbounded and removed with the washes? I think it is likely that by doing this, the MOI dropped drastically and that is the reason why they do not observe and IL-8 induction. Additionally, it is not clear how the ALI cultures were generated, if cells were fully differenticated, if TEER measurements wer performed, etc.

Reviewer 3 Report

The authors have undertaken interaction studies with pAEC from young children with and without CF and Aspergillus fumigatus. The use of pAEC is of high value and will add to our understanding derived from using epithelial cell lines and interaction with A. fumigatus. Nevertheless, the methodology and reporting of the results are showing concerning errors, which precludes any conclusions to be made.

Major comments:

The introduction should be rewritten and focus on Aspergillus – epithelial interactions as this the focus of the manuscript.

The susceptibility testing employed is not the one recommended for moulds. Microdilution methology should be performed. ECOFF for AmphoB for WT A. fumigatus is 1 not 2. A MIC of 2 for A. fumigatus is considered to be resistant, which would be highly remarkable. It makes no sense to provide MICs for fluconazole and 5-FC.

The reason for dividing the isolates based on ITS cluster is not clear and seems not to have added value to the observations.

Background of the ‘invasive strain’ should be provided (e.g. from neutropenic patient with IA?)

The authors should indicate if informed consent was obtained and if the study was approved by the ethics committee.

Explain why a MOI of 0.01 and 1 was used? Any titration done?

More details should be provided about the culture of the pAEC. To refer to another paper is not sufficient here.

Figure 1 is confusing, there should be no lines between the dots as these are individual measurements for one given condition. Number of experiments and replicates are missing. I would recommend to group the CF isolates vs the invasive isolates vs the ATCC strain.

In line 171, the authors are referring to an antifungal resistant isolate, but as none of the isolates indicate azole resistance and all the isolates are ampho B resistance, this is not correct.

Figure 2; how was hyphal growth quantified?

Figure 3 is confusing, there should be no lines between the dots as these are individual measurements for one given condition. To define a positive GM index as being > 1.5 doesn’t make sense in my opinion. There is increasing release of GM during culture in vivo, and looking at the pictures, there was in this experiment abundant growth of Aspergillus. Not sure if I understand where the authors see a bimodal distribution for IL-8 and IL-6 production as these are single timepoint experiments.

Figure 4 A+B is confusing, there should be no lines between the dots as these are individual measurements for one given condition. It is remarkable that there is no fungus visible in the sections D, E and G, H.

The discussion should focus on the Aspergillus-epithelial interactions observed.

The statement in line 259-260 can not be derived from the study presented.

The ampho B resistance amongst the strains used in this study needs a careful interpretation as this is very unusual. Proper susceptibility methodology should be used to verify these results.

The statement in line 329-332 and 339-342 can not be made based on the results presented.

Reference strains are lab adapted strain and it is questionable if they provide a same phenotype as the strains infecting patients.

Line 405, after vortexing conidial suspensions, the conidia are definitely in the lower half of the solution and not the upper half of the solution.